# Male coloration affects female gestation period and timing of fertilization in the guppy (*Poecilia reticulata*)

**Aya Sato**[1]\*, **Ryu-ichi Aihara**[2¤], **Kenji Karino**[2]

1 Department of Science Education, Cooperative Faculty of Education, Gunma University, Maebashi, Japan,
2 Department of Biology, Tokyo Gakugei University, Tokyo, Japan

¤ Current address: Sunagawa High School, Tokyo, Japan
\* satoaya@gunma-u.ac.jp

**Data Availability Statement:** All relevant data are within the paper and its S1 File.

**Funding:** The authors received no specific funding for this work.

## Abstract

The trade-up hypothesis is a female behavioral strategy related to mating with multiple males. In this hypothesis, females can produce high-quality offspring while avoiding the risk of losing reproductive opportunities by non-selective mating with males at first mating and then re-mating with more attractive males. As an internal mechanism to realize this behavioral strategy, we predicted that females would immediately fertilize their eggs when they mated with attractive males, whereas females would delay fertilization when they mated with unattractive males to trade-up sires of offspring. The guppy (*Poecilia reticulata*) is an ovoviviparous fish with internal fertilization, and females show a clear mate preference based on the area of orange coloration on the bodies of males. In addition, it is known that females show a re-mating strategy consistent with the trade-up hypothesis. We tested whether the attractiveness of mated males affected the gestation period and the timing of fertilization. Females were paired with either colorful males or drab males, and the gestation periods (the number of days from mating to parturition) were compared. In addition, we dissected the abdomens of the females at intervals of several days after mating and observed whether the eggs were fertilized. The gestation period in females that were paired with attractive colorful males was significantly shorter than that in females that were paired with drab males. We found that females that mated with colorful males also had their eggs fertilized earlier than those that mated with drab males. Our findings show that differences in the timing of fertilization according to attractiveness of the mate increase the opportunity for cryptic female choice and trading up.

## Introduction

Males of a species are not all equally successful in mating with females, and all males that succeed in mating do not equally sire offspring when a female mates with multiple males. Male-male competition and female mate choice, which compose sexual selection, have the potential to continue during post-copulation periods (following intromission or spawning) in the form

**Competing interests:** The authors have declared that no competing interests exist.

of sperm competition and cryptic female choice [1, 2]. Sperm competition is the post-copulatory equivalent of male-male competition and occurs when the ejaculates of two or more males compete for fertilization of a given set of ova [3]. Cryptic female choice corresponds to the post-copulatory female mate choice and occurs when females bias sperm use in favor of particular males [4, 5]. Therefore, males that succeed in sperm competition or are favored in cryptic female choice (or both) are able to sire more offspring among the multiple males that mated with a female. Eberhard [5] described a list of various female-controlled processes and activities during cryptic female choice, which modify the chance that a given copulation will result in offspring. To bias a male to sire offspring among multiple mated males, females can control not only the transport, storage, and use of sperm from the mated males, but also subsequently mate with other males [5].

The trade-up hypothesis is a female behavioral strategy involving multiple mating. When females choose mates at pre-copulation, they must take decisions to mate with a current male or seek a subsequent male. If females avoid mating with the current male to seek a more attractive male, it is possible that they might not find subsequent males and might miss the opportunity for reproduction. On the contrary, if females mate with the current male, they might miss the opportunity to find and mate with subsequent males of higher quality. The trade-up hypothesis describes a behavioral strategy in which females can maximize the genetic quality of their offspring when they encounter multiple males sequentially [6, 7]. The hypothesis suggests that in non-resource-based mating systems, with some degree of last-male sperm precedence, a female should mate with the first male that she encounters to ensure fertilization of eggs, but subsequently mate preferentially with males of higher genetic quality. Sequential mate choice in female newts *Triturus vulgaris* [8] and crickets *Gryllus bimaculatus* [9] have supported this hypothesis. Thus, females are relatively indiscriminate at the first mating, but they become increasingly choosy with each successive mating opportunity.

The guppy (*Poecilia reticulata*) is an ovoviviparous fish with internal fertilization and a non-resource-based promiscuous mating system [10, 11]. Female guppies show a clear mate preference in the pre-copulation phase: females in many populations prefer males with large orange areas on their bodies [12–14]. Females mate with multiple males and give birth to a brood with multiple paternity [15, 16]. Pitcher et al. [17] provided evidence that female guppies have a mating strategy that follows the trade-up hypothesis. They presented virgin females sequentially with two males of varying attractiveness, based on the amount of orange coloration on their bodies. They showed that female responsiveness to males presented second (second males) increased when the attractiveness of second males were greater than those of males presented first (first males), compared to when the attractiveness of second males were less than the first males or were the same as the first males. In addition, there was an overall tendency for a last-male advantage in paternity, and this advantage was more exaggerated when the second male was more ornamented than the first. Thus, female guppies may be able to maximize the genetic quality of their offspring through such a trade up of mated males during successive matings. While Pitcher et al. [17] demonstrated the trade-up hypothesis in the guppy from behavioral and paternity data, the cryptic processes occurring in the female internally remain unknown.

The cycle of egg production in guppies is non-superfetatious, that is, females give birth to a brood before the next ova are fertilized, and they carry a single brood at one time [11, 18]. During development, embryos are nourished by the yolk that females deposit in the eggs prior to fertilization [10]. Fertilization in Poeciliids occurs over a period of 1–5 d, and after 1–8 d postparturition [18]. The reproductive cycle of guppies typically lasts approximately 25–30 d, although considerable variation has been observed [11]. For example, the gestation period (days from mating to parturition) of virgin females can vary anywhere from 20 to 65 d [19,

20]. The gestation period is composed of two phases: the interval between insemination and fertilization (which may also include a yolk-loading phase) and the ensuing interval between fertilization and parturition [20]. Evans and Magurran [19] showed that the gestation period in female guppies paired with multiple males was shorter than those mated with a single male, while females that paired with a single male were likely to delay the process of fertilization for a more suitable mating. In addition, the gestation period has been shown to be shorter for female guppies that mated with colorful males than for those that mated with drab males [21]. Considering the above findings, we predict that females that mate with more ornamented males will immediately fertilize their ova, whereas females that mate with less ornamented males will delay fertilization. In this study, we directly tested whether the sexual attractiveness of mated males affects the timing of fertilization in female guppies. We observed whether the ova were fertilized by opening their abdomens between 2 and 22 d after pairing with colorful or drab males. Additionally, we measured the gestation period (duration from mating to parturition) between females that mated with colorful males and those that mated with drab males, and compared whether the difference in the timing of fertilization according to male coloration reflected the difference in the gestation period.

## Materials and methods

### Study subject

The fish used in this study were laboratory-reared descendants of feral guppies from the Hiji River (26˚ 43 N, 128˚ 11 E), Okinawa Island, southern Japan. Females in this population prefer males with larger orange areas on the body than those with smaller ones [14, 22]. Fish were reared in a laboratory at Gunma University. Water temperature was maintained at 27 ± 2˚C. Illumination was provided by fluorescent tubes, and the light/dark regime was set at 12/12 h. All fish were fed once daily with newly hatched brine shrimp and commercial flake food (Tetramin; Tetra Werke, Melle, Germany).

Males were reared in 40 L mixed-sex aquaria (45 × 30 × 30 cm) containing approximately 50 males and 50 females. To eliminate the influence of the previous mating, we used virgin females that were reared in 12 L single-sex aquaria (37 × 22 × 25 cm) from 30 to 60 d after birth.

### Ethics statement

Female laparotomy treatments were approved by the Animal Care and Experimentation Committee of Gunma University, Japan (permit numbers: 20–040). All experiments were conducted according to the guidelines of the committee.

### Test fish

We used mature individuals (> 5 months old) as the test fish. Before the start of the experiment, test fish were anesthetized with 2-phenoxyethanol solution, and body size (i.e., total length from snout to the tip of the caudal fin and standard length from snout to root of the caudal fin) was recorded to the nearest 0.1 mm by using a vernier caliper.

To quantify body coloration, males were photographed on the left and right sides following body size measurements. For photography, a digital camera (TG-3; Olympus, Tokyo, Japan) was placed at a distance of 4 cm above the male in a Petri dish containing 2-phenoxyethanol solution for anesthetization. Two lights (krypton bulb 25 W, daylight color, Sunlamp KR R45, Asahi, Tokyo, Japan) illuminated both sides of the petri dish from a position 20-cm away from the fish. The male body was fully immersed in the anesthetic solution; after a complete lack of

movement, the male body was gently shaken in the solution to open the caudal fin. We took some photographs and selected clearer photographs and those in which the caudal fin was fully opened for analysis. The selected images were analyzed using Adobe Photoshop Elements 14 (Adobe Systems, San Jose, CA, USA). The relative area of the orange spots was calculated as the ratio of the total area of orange spots to the total area of the body and caudal fin. We averaged the relative area of the orange spots from the right and left sides and considered it as the orange spot area of the male.

For pairing, a tetrad was created with two virgin females and two males. The two females in a tetrad had similar body sizes (< 2-mm difference in total length). The two males in a tetrad had similar body sizes (< 2-mm difference in total length), while the orange spot areas differed. We assigned individuals with a relatively larger orange spot area as colorful males and individuals with a relatively smaller orange spot area as drab males. Before pairing, the two females in a tetrad were placed in a small transparent tank (11 × 11 × 9 cm) that was subsequently placed in a male rearing aquarium (containing approximately 50 adult males) for 24 h, in order to enhance mate choosiness of the female by making visual contact with various males.

## Pairing

The pairing tank was a transparent plastic aquarium (18 × 10 × 11 cm) and was divided into two compartments (9 × 10 × 11 cm) using a transparent divider. The tank was filled with water to a depth of 8 cm, and the bottom was covered with gravel. The water temperature and light conditions were the same as those mentioned previously. One tetrad was placed in the pairing tank, with the two males in one compartment of the tank and two females in the other compartment during the acclimation period. After a 24 h acclimation period, one male and one female in each compartment were switched such that one compartment contained a colorful male and a female, and the other compartment contained a drab male and a female. Each male-female pair freely mated for 24 h, which was considered as the "mating period".

After pairing, the test males were individually housed for use again as a different tetrad or removed to a stock aquarium. The females were individually reared after pairing in parturition tanks (10 × 15 × 9 cm) containing 1.2 L of water and gravel. The females were allocated to either parturition treatments or laparotomy treatments; two females in the same tetrad were allocated to the same treatment. As a shelter for the offspring, an artificial aquatic plant made of thin green polyethylene tape was placed in the tanks containing females in the parturition treatments. These tanks were observed daily and checked for newborn broods. When the females gave birth, we calculated the number of days from pairing to parturition.

## Female laparotomies and discrimination of fertilization

The abdomens of females in the laparotomy treatment were opened between 2 and 22 d after pairing, and the ova or embryos were extracted. We set a maximum of 22 d after pairing because the shortest gestation period was 25 d in the above parturition treatment. The two females from the same tetrad were laparotomized on the same day. When the same pair of colorful and drab males were used for different tetrads, the females of different tetrads were laparotomized on different days after pairing. In addition, we also laparotomized nine virgin females as a control treatment.

After the females were anaesthetized in 0.7 mL/L 2-phenoxyethanol solution, they were euthanized by over-anesthetization using 2-phenoxyethanol. Using scissors and fine forceps, the ventral side of the abdomen was dissected from the cloaca to the anterior region. The ovaries were extracted under a dissecting microscope and placed in a Petri dish containing fresh

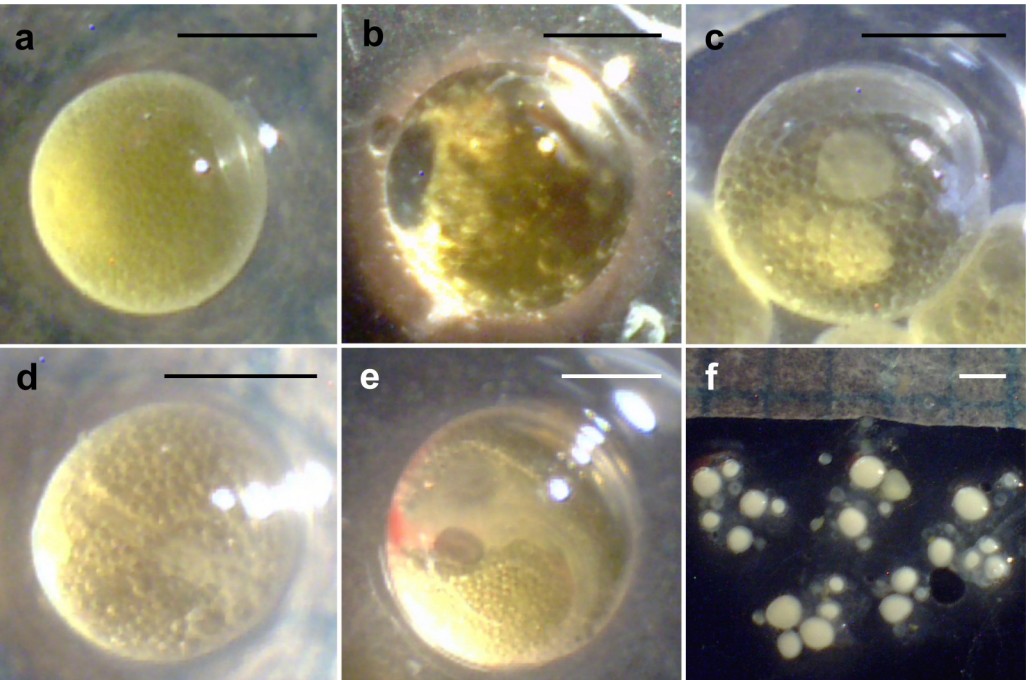

**Fig 1. Unfertilized and fertilized ova.** a) early yolked ovum, b) mature ovum, c) blastodisc embryos, d) primitive streak embryo, e) developing embryo, and f) immature ova. Scale bar = 1 mm.

water. The thin connective tissue of the ovary was carefully incised to isolate individual ova and embryos using fine tweezers. The ova and embryos were examined under a dissecting microscope to determine their developmental stages.

The ova and embryonic developmental stages were determined according to Haynes [23] and Martyn et al. [24]. When females held early yolked ovum (Fig 1A) or mature ova that were fully yolked (Fig 1B), we considered their ova to be unfertilized. When females had blastodisc embryos (Fig 1C), primitive streak embryos (Fig 1D), or embryos after these stages (Fig 1E), we considered their ova to be fertilized. If females had only small white immature ova (Fig 1F), we excluded them from the analyses as these ova were considered not ready for fertilization.

## Statistical analyses

A generalized linear mixed model (GLMM) was used to compare gestation period between females paired with colorful males and those paired with drab males. We used the gestation period as a dependent variable, the male coloration (colorful/drab), brood size, and female standard length as fixed factors, and tetrad identity as a random factor. After the dependent variable was log-transformed (Kolmogorov-Smirnov test in post-transform; $D = 0.130$, df = 46, $P = 0.050$) to meet the model assumptions of normality, the data were fitted to a normal distribution with an identity link function.

To examine whether the coloration of the males paired with affected the timing of fertilization, we conducted GLMM using binomial data. Values of 0 and 1 were assigned to females that had unfertilized and fertilized ova, respectively, and the binary data were considered as a dependent variable. We used male coloration (colorful/drab) and the number of days from pairing as fixed factors. Tetrad and male identities were entered into the models as random factors, because some males were used in several different tetrads. The data were fitted to a

binomial distribution using a logit link function. In addition, it was estimated that the inflection points of fertilization occurred in 50% of females that were paired with either colorful or drab males. We performed logistic regression analyses and calculated the inflection points from the regression equation as Y = 1/exp [−(α + β X)]. All statistical tests were performed using SPSS version 26.0.0.0 and R version 3.3.3.

## Results

We created 60 pairs of 54 colorful males and 51 drab males. Seventy-nine tetrads were created from 60 male pairs and 158 females. The body size of colorful males was 16.4 ± 1.2 mm (standard length; mean ± standard deviation) and that of drab males was 16.4 ± 1.2 mm (paired t-test; n = 60 pairs, t = 0.312, p = 0.714). The body size of females that paired with colorful males was 21.8 ± 1.9 mm and the body size of females that paired with drab males was 21.8 ± 1.9 mm (paired t-test; n = 79 pairs, t = -0.078, p = 0.938). The orange area of colorful males was 10.9% ± 4.7%, the orange area of drab males was 2.9% ± 1.2%, and the difference in the orange area between colorful and drab male pairs was 7.9% ± 5.0% (paired t-test; n = 60 pairs, t = 12.241, p < 0.001). We assigned the parturition treatment to females in 24 tetrads and the laparotomy treatment to females in 55 tetrads. All relevant data are available in the manuscript and S1 File.

In the parturition treatment, one female that paired with a colorful male and another that paired with a drab male did not give birth to a brood even 120 days after pairing. These data were excluded from the analyses. The results of the GLMM showed that the body size of females and brood size did not have a significant effect; therefore, these variables were removed from the model. Finally, male coloration significantly affected gestation period (Table 1). The gestation period in females that were paired with colorful males (mean ± standard deviation: 31.1 ± 7.3 d) was significantly shorter than that in females paired with drab males (39.0 ± 9.8 d; Fig 2). Brood size was not affected by male coloration, although it was significantly affected by the body size of females (Table 1).

The virgin females had immature ova, early yolked, or mature ova (one female: only immature ova; two females: only early yolked ova; four females: only mature ova; two females: early yolked and mature ova). In the laparotomy treatment, all three females that paired with a certain colorful male had early yolk ova when laparotomy was performed (8, 15, and 22 d after pairing). Similarly, all three females that paired with a certain drab male had early yolked ova when laparotomy was performed (6, 12, and 20 d after pairing). These data were excluded from the analyses because those males were likely to be infertile. In addition, four females (two colorful pairs and two drab pairs) died before laparotomy treatment. Eight females (four colorful pairs and four drab pairs) had only small white immature ova or no ova at all. These data were excluded from the analyses. Finally, the number of females used in the analysis was 46 females paired with colorful males and 46 females paired with drab males.

The number of days from pairing and male coloration (colorful/drab) significantly affected the timing of fertilization (Table 2, Fig 3). The inflection points at which half the females had fertilized ova were 11.7 d for colorful pairs and 18.2 d for drab pairs.

**Table 1. Results of the generalized linear mixed model in parturition treatment.**

|  | Estimate ± Standard error | t | P |
|---|---|---|---|
| Gestation period |  |  |  |
| Male type (colorful / drab) | 0.21 ± 0.10 | 3.72 | <0.001 |
| Brood size |  |  |  |
| Male type (colorful / drab) | 0.91 ± 1.23 | 0.75 | 0.454 |
| Female size | 1.30 ± 0.42 | 3.09 | 0.002 |

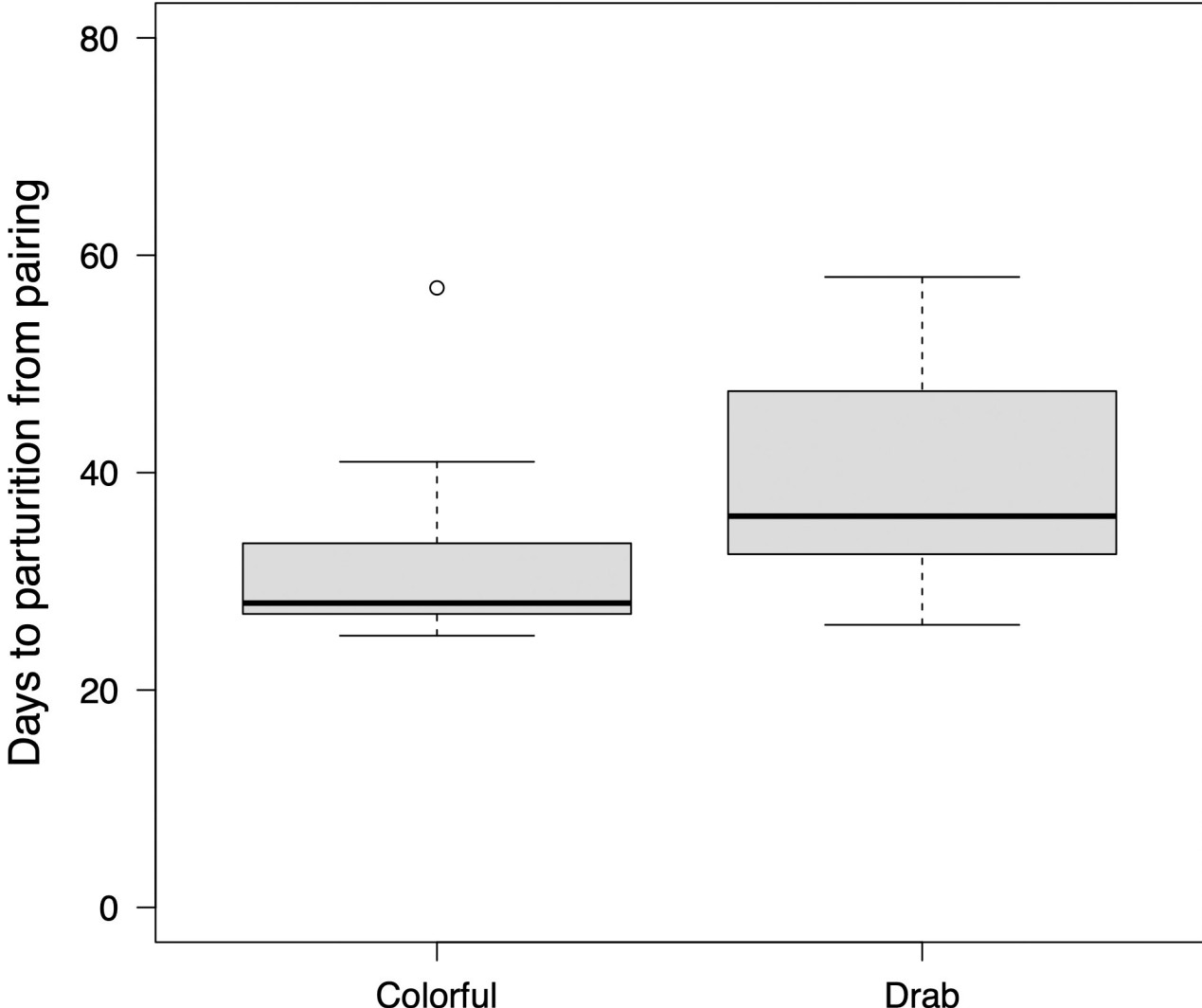

**Fig 2. Male coloration and female gestation period.** Boxplots of gestation period when females paired with colorful males (sample size n = 23) and females paired with drab males (n = 23). The gestation period indicates values before transformation.

## Discussion

We investigated whether the male orange spot area (an indicator of male sexual attractiveness) affected the timing of female fertilization. The gestation periods of females that mated with males possessing larger orange areas were significantly shorter than those of females that mated with males possessing smaller orange areas. The gestation period in the guppy broadly comprises two phases: (1) the interval between insemination and fertilization (which may also include a yolk-loading phase), and (2) the ensuing interval between fertilization and

**Table 2. Results of the generalized linear mixed model in laparotomy experiment.**

|  | Estimate ± Standard error | z | P |
|---|---|---|---|
| Male type | −1.08 ± 0.51 | −2.10 | 0.035 |
| Days | 0.21 ± 0.05 | 4.24 | <0.001 |

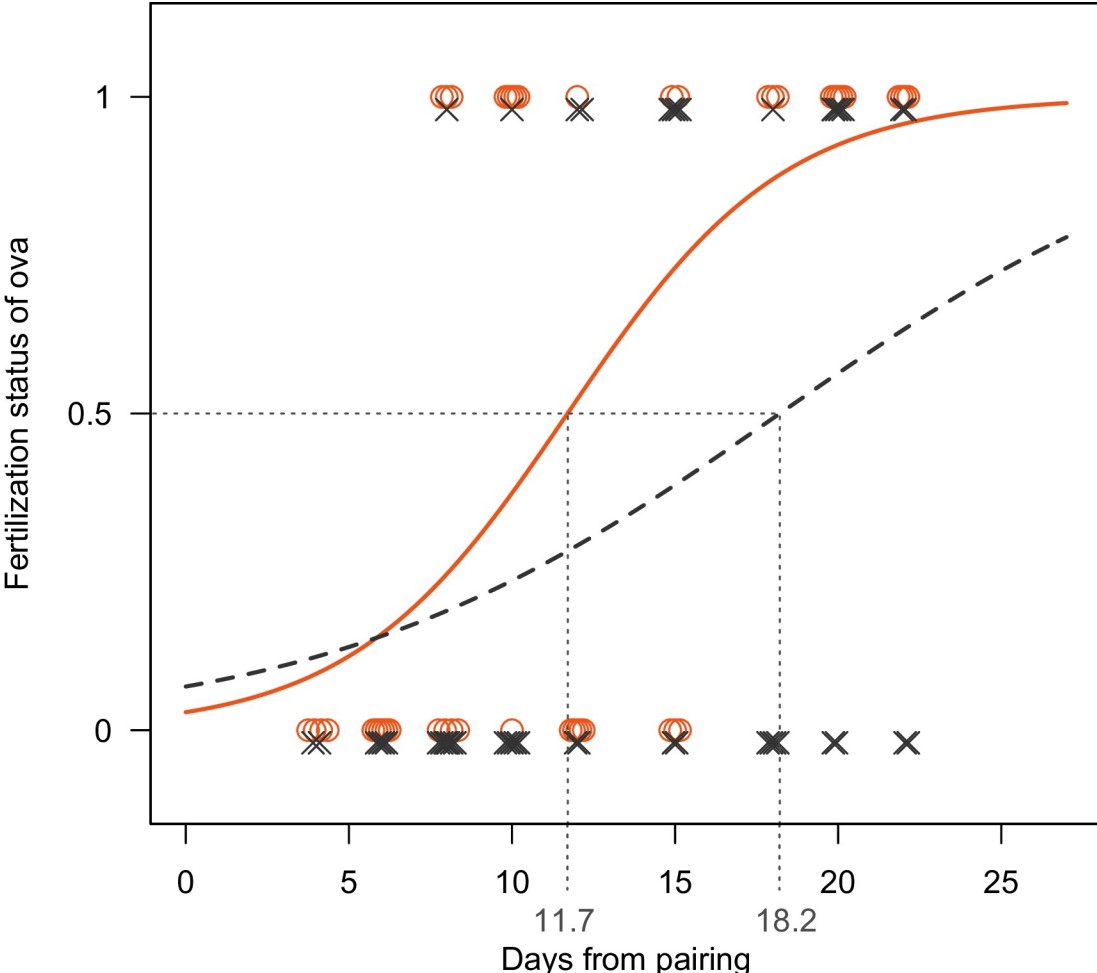

**Fig 3. Male coloration and timing of fertilization.** Circles indicate females that paired with colorful males and crosses indicate females that paired with drab males. Solid and dotted curves indicate logistic regression lines of females that paired with colorful and drab males, respectively.

parturition [20]. The timing of fertilization in females that paired with males possessing larger orange areas was significantly earlier than that in females that paired with males possessing smaller orange areas. Interestingly, the difference in mean gestation period between females that paired with colorful males and those that paired with drab males was 7.9 d, while the difference in inflection points that 50% of females fertilized their ova was 6.5 d, that is, these differences were approximately similar. These results suggest that the difference in gestation periods depending on mate ornamentation is possibly caused by the difference in timing of fertilization.

In the laparotomy treatments, some females had unfertilized ova when they were paired with drab males more than 18 days after pairing, whereas all females had fertilized ova when they were paired with colorful males. Therefore, it is possible that the statistical analysis may have overestimated the effect of male coloration on the timing of fertilization because it is possible that some females that paired with drab males failed to fertilize their ova. For example, the drab males possibly failed to copulate or deliver sperm during copulation, or the sperm of drab males possibly failed to fertilize the ova. It has previously been reported that male guppies with small orange areas transfer less sperm to females than those with large orange areas [25];

this is the result of female control [26]. However, in the parturition treatment in this study, the fact that the number of females that did not give birth was the same regardless of mate coloration, rejects the possibility that the females that paired with drab males failed to mate or receive sperm. In the parturition treatment, the shortest gestation period was 25 d (this is regarded as the minimum time required for embryonic development to parturition from fertilization), whereas the longest gestation period was 58 d. Therefore, it is possible that the ova could have been fertilized more than 18 d after pairing, because the gestation duration would be 44 d (i.e., 25 d after fertilization), if the ova were fertilized at 19 d after pairing.

We assumed that the difference in the timing of fertilization depending on male sexual ornamentation is a female control that enables a trade-up strategy with multiple mating. However, differences in the timing of fertilization may also be caused by males. A positive relationship between carotenoid-based male ornamentation and sperm quality (velocity) has been reported in several species [27–29]. In fact, it is known that colorful male guppies produce faster sperm [30, 31] (but see [32, 33]). In addition, not only sperm quality but also the number of sperm may affect the timing of fertilization. Colorful male guppies have been reported to produce more sperm [34, 35] and transfer more sperm to females than drab males during cooperational copulations [25]. Alternatively, male semen products, such as induction of ovulation or oviposition and resistance to further insemination by other males [36, 37], may affect the timing of fertilization. Effects on oviposition and resistance to mating are particularly well-documented and widespread in insects and ticks [5], although it has not been reported in guppies. The quality or number of sperm or semen products may affect the timing of fertilization. However, in guppies, it has been reported that the male orange area does not affect the gestation period when females produce their brood under conditions excluding female control, such as artificial insemination [38, 39]. In addition, it has been reported that artificially adjusted ejaculate size does not influence the female gestation period [39]. This supports our assumption that the difference in the timing of fertilization depending on male sexual ornamentation is an effect of cryptic female choice.

Control of the timing of fertilization (or oviposition after copulation) by females has been reported in several arthropods [40–43]. In these species, the lengthened period of female sexual receptivity due to delayed fertilization increases the opportunity for re-mating; as a result, females are able to bias fertilization towards preferred males [41]. By mating with colorful males with large orange areas, female guppies can gain indirect benefits that they produce offspring with high anti-predator ability [38] and algal-foraging ability [44], and sons with more attractive ornaments inherited from the sire [45, 46]. Virgin females are less choosy when they mate for the first time than in subsequent matings, and their probability of re-mating increases if the second male is more attractive than the first [17]. When females mate sequentially with two males in accordance with their mate choice and decision of re-mating, paternity is usually biased towards the second male [17, 47–49]. However, Magris et al. [50] reported that when behavioral interactions between males and females and potential differences in ejaculate size between males were removed using artificial insemination, the second male precedence detected after natural copulation was reversed. Accordingly, they concluded that the last male precedence observed after natural copulations ascribes to female control of the number of sperm transferred during copulation. Thus, through female guppy control of re-mating and the number of sperm transferred during copulation, female guppies can selectively produce higher-quality offspring while avoiding the risk of losing reproductive opportunities. The results of the present study indicate that differences in the timing of fertilization according to male attractiveness of the mate also increase the opportunity for cryptic female choice and trading up.

Females that mate at first with unattractive males will gain the benefit of having offspring of high quality sired by their second attractive mates by delaying fertilization. However, if females do not encounter more attractive males despite delaying fertilization, they will lose future opportunities for reproduction. How long after the first mating will the cost of delaying fertilization be higher than the benefits from trade-up? The duration of gestation in this study was 25–58 d. The duration of gestation in other studies was approximately 22–65 d [19] and 20–60 d [20]. The maximum number of days is approximately twice that of the typical reproductive cycle (approximately 25–30 d [11]). Therefore, the loss of opportunity for one reproductive cycle may be a relatively large cost for females. The cost of delayed fertilization varies according to the expectation of remating with more attractive males or the remaining opportunity for reproduction. Further studies investigating the relationship between these factors and delayed fertilization will contribute to the understanding of the female trade-up strategy.

## Conclusions

We found that the timing of fertilization in female guppies was significantly affected by male attractiveness, and females that mated with drab males fertilized later than those that mated with colorful males. This result supports our hypothesis that females that mate with attractive males at the beginning of the sexually receptive period and expected high-quality offspring, fertilize their eggs immediately after mating, whereas the females that mate with less attractive males, delay fertilization for trade-up by sperm from more attractive males. In the guppies, it is known that females re-mate with more colorful males than males that mated previously, and the paternity is biased to the last males [17]. The results of the present study indicate that female guppies increase the opportunity for re-mating and trading up by delaying the timing of fertilization.

## Supporting information

**S1 File. Raw data of experiments.**
(XLSX)

## Acknowledgments

We thank Kaoru Yoshida for teaching us the laparotomy technique. We are grateful to Aki Mieno, Ayaka Niwa, Mizuki Ogata, and Yusuke Sakai for their help with animal care. We thank Dr. Kausalya Shenoy, Dr. Hon Jung Liew, and two anonymous reviewers for their comments and suggestions on an earlier version of the manuscript.

## Author Contributions

**Conceptualization:** Aya Sato.

**Data curation:** Aya Sato, Ryu-ichi Aihara.

**Formal analysis:** Aya Sato.

**Investigation:** Aya Sato, Ryu-ichi Aihara.

**Resources:** Ryu-ichi Aihara, Kenji Karino.

**Visualization:** Aya Sato.

**Writing – original draft:** Aya Sato.

**Writing – review & editing:** Kenji Karino.

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
