## [Decision Letter · Decision Letter 0]

29 Jun 2021

PONE-D-21-14329

Male coloration affects female gestation period and timing of fertilization in the guppy (Poecilia reticulata)

PLOS ONE

Dear Dr. Sato,

Thank you for submitting your manuscript to PLOS ONE. After careful consideration, we feel that it has merit but does not fully meet PLOS ONE’s publication criteria as it currently stands. Therefore, we invite you to submit a revised version of the manuscript that addresses the points raised during the review process.

We look forward to receiving your revised manuscript.

Kind regards,

Eelke Snoeren

Academic Editor

PLOS ONE

Journal Requirements:

Reviewers' comments:

Reviewer's Responses to Questions

**Comments to the Author**

1. Is the manuscript technically sound, and do the data support the conclusions?

Reviewer #1: Yes

Reviewer #2: Yes

2. Has the statistical analysis been performed appropriately and rigorously? 

Reviewer #1: No

Reviewer #2: Yes

3. Have the authors made all data underlying the findings in their manuscript fully available?

Reviewer #1: Yes

Reviewer #2: Yes

4. Is the manuscript presented in an intelligible fashion and written in standard English?

Reviewer #1: No

Reviewer #2: Yes

5. Review Comments to the Author

Reviewer #1: This is an interesting and well-conceived study aimed at testing the prediction that female guppies delay egg maturation and fertilization when mated with a less colourful male.

The results seems to support this hypothesis, as the time between mating and parturition is longer in the female mated with drab males as compared to those mated with colourful males. Through dissections, the authors demonstrated that egg maturation proceeds faster in the colourful group.

I think that this may be a valuable contribution, but I have also a few suggestions that may be useful to revise the MS.

English requires extensive editing, and I suggest a revision from a native English speaker.

A general comment: as you also reckon, it is difficult to distinguish between male-controlled and female-controlled effects of mating on fertilization. It may be that the quantity of ejaculate transferred during mating affects subsequent ovulation processes. Previous studies that used artificial insemination failed to find an effect of male colouration, but equal number of bundles were used. It may be interesting to test if different number of sperm inseminated affects gestation length.

Statistical methods:

1) It is not clear if gestation data were tested for normality. From a visual inspection it seems that a log transformation should be applied.

2) I do not see any good reason to use GEE. Data from tetrads in the parturition exp. are not repeated measures, rather they are not independent, and I think a generalized linear mixed model should be more appropriate. Male ID and tetrad can be entered as random factors, and male colour as fixed factor. Probably results will not change much.

3) The use of GEE may be more appropriate for the exp. on egg maturation, although here also a more commonly used generalized LMM may be used too. Indeed, it is not the same male that it is measured at different times, but different females mated with the same male (i.e. not independent) dissected once at different times after mating.

Other points:

Table 1. You should check if male type remains significant after removing for non-significant covariates (brood size and female size).

Line 225: please also give the mean difference +/- SD between that two males

Line 315: there is a study demonstrating a first male fertilization success when female influenced is controlled using artificial insemination (Magris et al. 2017 Animal Behaviour 131:45-55).

Line 318: this is not exact. Your results demonstrate that there is a delay in fertilization over days. Experiments on trade-up (e.g. Evans et al. 2001, Pitcher et al. 2003) involved subsequent matings occurring over a much shorter time (1h up to 24 h). While the observed delay in ovulation mat increase the temporal window within which female cryptic choice can occur, your data do not directly demonstrate that this is involved in trading up. So your conclusion is that delayed fertilization increase the opportunity for cryptic female choice and trading up, not that it is involved in trading up, which is most probably determined by females controlling the number of sperm transferred during matings.

Line 328: I like this argument, this is an interesting point.

Reviewer #2: Dear Authors,

This is a well designed study, and the findings are useful to the scientific community. I have one major change that I would like to see in Figure 2. All edits and comments are embedded within the attached PDF file of your manuscript.

Best wishes!

6. PLOS authors have the option to publish the peer review history of their article (what does this mean?). If published, this will include your full peer review and any attached files.

Reviewer #1: No

Reviewer #2: **Yes: **Kausalya Shenoy

---

## [Author Response · Author response to Decision Letter 0]

6 Aug 2021

We would like to express our appreciation for the reviewers’ suggestions for revising our manuscript. Addressing their comments has significantly improved the manuscript. The line numbers refer to those of the revised manuscript unless otherwise noted, in which the corresponding changes have been highlighted in the revised manuscript with changes tracked.

“When submitting your revision, we need you to address these additional requirements.

1. Please ensure that your manuscript meets PLOS ONE's style requirements, including those for file naming.”

>>We ensured that revised manuscript meets PLOS ONE’s style requirement.

Reviewers' comments:

Reviewer's Responses to Questions

Comments to the Author

“2. Has the statistical analysis been performed appropriately and rigorously?

Reviewer #1: No”

>>We have revised the statistical analysis according to the comments by reviewer #1. Details are described later. 

“4. Is the manuscript presented in an intelligible fashion and written in standard English?

Reviewer #1: No”

>> The revised manuscript has been edited by a professional language editing service (highlighted in gray). We have attached a Certificate of English Editing. 

5. Review Comments to the Author

“Reviewer #1: This is an interesting and well-conceived study aimed at testing the prediction that female guppies delay egg maturation and fertilization when mated with a less colourful male.

The results seems to support this hypothesis, as the time between mating and parturition is longer in the female mated with drab males as compared to those mated with colourful males. Through dissections, the authors demonstrated that egg maturation proceeds faster in the colourful group.

I think that this may be a valuable contribution, but I have also a few suggestions that may be useful to revise the MS.

English requires extensive editing, and I suggest a revision from a native English speaker.”

>> We appreciate the comment. According to the comments by reviewer #1, the revised manuscript has been edited by a professional language editing service (highlighted in gray). 

A general comment: 

“as you also reckon, it is difficult to distinguish between male-controlled and female-controlled effects of mating on fertilization. It may be that the quantity of ejaculate transferred during mating affects subsequent ovulation processes. Previous studies that used artificial insemination failed to find an effect of male colouration, but equal number of bundles were used. It may be interesting to test if different number of sperm inseminated affects gestation length.”

>>Thank you for the comment. Evans & Magurran (2000, [19] in the References list) showed that females that mated multiple males had a shorter gestation period than females that mated with a single male. From this comment of reviewer #1, we expect that the findings of Evans & Magurran (2000) can also be explained by the quantity of ejaculate transferred during mating. On the other hand, Pilastro et al. (2008) reported that adjusted ejaculate quantities did not affect the gestation period in artificial insemination. Therefore, we have added a discussion on the possible effects of the different quantities of sperm between colorful and drab males on the female gestation period (Lines 305-308, 315-316).

Statistical methods:

“1) It is not clear if gestation data were tested for normality. From a visual inspection it seems that a log transformation should be applied.”

>> Based on the result of the Kolmogorov-Smirnov test with Lilliefors significance correction, the gestation period data were not significantly normally distributed (D=0.175, df=46, p=0.001). Therefore, we log transformed the gestation period data (KS test in post-transform; D=0.130, df=46, p=0.050) and applied Box-Cox transformation (D=0.129, df=46, p=0.054). Because the log-transformed values fit the model better in the following test, we used log-transformed values for the analysis. We added the information that the gestation period was log-transformed (Lines 205-207).

“2) I do not see any good reason to use GEE. Data from tetrads in the parturition exp. are not repeated measures, rather they are not independent, and I think a generalized linear mixed model should be more appropriate. Male ID and tetrad can be entered as random factors, and male colour as fixed factor. Probably results will not change much.”

>> Thank you for your helpful suggestion. We have now used a generalized linear mixed model for the analysis of gestation data in the revised manuscript. We used the log-transformed gestation period as a dependent variable, the male coloration (colorful/drab) as a fixed factor, and the tetrad ID as a random factor. As suggested by reviewer #1, males were not used repeatedly in the parturition experiment. Therefore, we excluded male ID from the model. In addition, female SL and brood size were excluded from the model, according to the comment in the “other points” section of reviewer #1 and the comment by reviewer #2. The results did not change significantly. We have revised the description of the “statistical analyses” (Lines 202-205) and “results” (Lines 235-237, Table 1). 

“3) The use of GEE may be more appropriate for the exp. on egg maturation, although here also a more commonly used generalized LMM may be used too. Indeed, it is not the same male that it is measured at different times, but different females mated with the same male (i.e. not independent) dissected once at different times after mating.”

>> We have used a generalized linear mixed model for analysis of the laparotomy experiment in the revised manuscript. We used tetrad and male ID as random factors. The results were the same as those when using GEE. We have accordingly revised the description of the “statistical analyses” (Lines 210, 212-214) and “results” (Table 2).

Other points:

“Table 1. You should check if male type remains significant after removing for non-significant covariates (brood size and female size).”

>> We removed the brood size and female size from the model. The effect of male type remained significant. We have revised the results (Lines 235-237) and Table 1 accordingly. 

“Line 225: please also give the mean difference +/- SD between that two males.”

>>We have added the mean difference+/- SD between that two males: “the difference in the orange area between colorful and drab males pairs was 7.9 % ± 5.0 %” (Lines 228-229).

“Line 315: there is a study demonstrating a first male fertilization success when female influenced is controlled using artificial insemination (Magris et al. 2017 Animal Behaviour 131:45-55).”

>> Thank you for your helpful suggestion. We have cited this reference and revised the sentence (Lines 328-332).

“Line 318: this is not exact. Your results demonstrate that there is a delay in fertilization over days. Experiments on trade-up (e.g. Evans et al. 2001, Pitcher et al. 2003) involved subsequent matings occurring over a much shorter time (1h up to 24 h). While the observed delay in ovulation mat increase the temporal window within which female cryptic choice can occur, your data do not directly demonstrate that this is involved in trading up. So your conclusion is that delayed fertilization increase the opportunity for cryptic female choice and trading up, not that it is involved in trading up, which is most probably determined by females controlling the number of sperm transferred during matings.”

>> Thank you for your helpful suggestions. We agree with this comment. We have revised this conclusion accordingly (Lines 334-335). 

“Line 328: I like this argument, this is an interesting point.”

>> Thank you. We also look forward to future research. 

Reviewer #2: Dear Authors,

“This is a well designed study, and the findings are useful to the scientific community. I have one major change that I would like to see in Figure 2. All edits and comments are embedded within the attached PDF file of your manuscript.

Best wishes!”

>> Thank you for the comments. We appreciate correcting the English. The language corrected according to the comments by reviewer #2 is highlighted in blue in the revised manuscript with changes tracked. We have revised the manuscript as follows in response to other comments. 

Lines100-102（previous manuscript）, Line 99-100（revised manuscript）

We received the following comments from the reviewer #2 to our sentence (“In this study, we investigated whether the sexual attractiveness of mated male would affect gestation period and timing of fertilization in female guppies.”)

: “didn’t Evans & Magurran alredady study this?”

>> We have responded as follows:

Evans & Magurran (2000, [19] in the References list) showed that females that mated with multiple males had a shorter gestation period than females that mated with a single male. And it was suggested that females that mated with single male may have delayed the process of fertilization. In this study, we directly tested whether male sexual attractiveness affected the timing of fertilization when a females mated with a single male, through laparotomy experiments.

Line118-122（previous manuscript）

Reviewer #2 commented on the paragraph of “Experimental design” as “Move to introduction after prediction statement”.

We have moved that paragraph to lines 101-105. 

L131-132（previous manuscript）

Reviewer #2 commented on the total and standard length as ”caudal fin included or not ?”.

We have added that the total length included the caudal fin and that standard length did not include the caudal fin (Lines 129-130). 

L136（previous manuscript）

Reviewer #2 commented on the light for photography as a ”type of bulb? Color temperature?”.

We have added the information that we used the krypton bulb of daylight (Sunlamp KR R45, Asahi). (Lines 135-136)

L140（previous manuscript）

Reviewer #2 commented regarding the male photography as “Did you ensure that caudal fin is fully opened to avoid errors in area measurement?”.

We have added the following to the section on male coloration measurement: 

“The male body was fully immersed in the anesthetic solution, and after complete lack of movement, the male body was gently shaken in the solution to open the caudal fin. We took some photographs and selected the clearer photographs, and the ones in which the caudal fin was fully opened for analysis (Lines 137-140)”

L147（previous manuscript）

Reviewer #2 commented on the procedure of visually contacting the test females with many males prior to pairing, as to whether all females were placed in a small tank or each female was placed in a small tank. 

We have added that the two females in a tetrad were placed in a small tank for visual contact between the females and the males (Line 150). 

Line 204（previous manuscript）

Reviewer #2 commented on the analysis of parturition experiment as:

The brood size and the female standard length can be taken out of the model if these effects were not significant. 

We changed the method of analysis from general estimating equations (GEE) to generalized linear mixed models (GLMM) according to the comment of reviewer #1. Moreover, the effects of brood size and female size were not significant. Therefore, we removed them from the final model (Lines 235-237, Table 1).

Line210-211（previous manuscript）

Reviewer #2 commented on the analysis of parturition experiment that tetrad and male ID can be taken out of the model if these effects were not significant. 

We have also changed this analysis from GEE to GLMM according to the comment of reviewer #1. Because we used tetrad and male ID as random factors in the GLMM, these factors were used to detect the relationship between the fixed variables and the dependent variable, but did not detect a direct effect on the dependent variable. Therefore, we left them in the model as random variables.

Table 1

According to the changes in the results of analyses, we revised Table 1. In addition, we revised Fig 1 according to the comment from reviewer #2. 

Line 250 （previous manuscript）

Reviewer #2 commented on the result that excluded some females from the analysis, as “Number of females remaining in laparotomy treatment was?”

We have added number of females remaining in the analysis of laparotomy treatment: 

“Finally, the numbers used in the analysis were 46 females that paired with colorful males and 46 females that paired with drab males” (Lines 257-258).

Fig. 2

Reviewer #2 commented that Fig.2 should change to likely a bar graph. Based on this comment, we have revised Fig. 2. Because original data of the gestation period were not normality, we have changed Fig.2 to boxplots. 

Fig.3

Reviewer #2 commented on Fig.2 as “Please make inflection points and the corresponding No of days from pairing for each regression line.”. 

We have accordingly revised Fig. 3.

---

## [Decision Letter · Decision Letter 1]

6 Oct 2021

PONE-D-21-14329R1Male coloration affects female gestation period and timing of fertilization in the guppy (*Poecilia reticulata*)***PLOS ONE*

*Dear Dr. Sato,*

*Thank you for submitting your manuscript to PLOS ONE. After careful consideration, we feel that it has merit but does not fully meet PLOS ONE’s publication criteria as it currently stands. Therefore, we invite you to submit a revised version of the manuscript that addresses the points raised during the review process.*

 *Please submit your revised manuscript by Nov 20 2021 11:59PM. If you will need more time than this to complete your revisions, please reply to this message or contact the journal office at plosone@plos.org. *

*Please include the following items when submitting your revised manuscript:*

*A rebuttal letter that responds to each point raised by the academic editor and reviewer(s). You should upload this letter as a separate file labeled 'Response to Reviewers'.*

*A marked-up copy of your manuscript that highlights changes made to the original version. You should upload this as a separate file labeled 'Revised Manuscript with Track Changes'.*

*An unmarked version of your revised paper without tracked changes. You should upload this as a separate file labeled 'Manuscript'.*

**

*If applicable, we recommend that you deposit your laboratory protocols in protocols.io to enhance the reproducibility of your results. Protocols.io assigns your protocol its own identifier (DOI) so that it can be cited independently in the future. For instructions see: https://journals.plos.org/plosone/s/submission-guidelines#loc-laboratory-protocols. Additionally, PLOS ONE offers an option for publishing peer-reviewed Lab Protocol articles, which describe protocols hosted on protocols.io. Read more information on sharing protocols at https://plos.org/protocols?utm_medium=editorial-email&utm_source=authorletters&utm_campaign=protocols.*

*We look forward to receiving your revised manuscript.*

*Kind regards,*

*Khor Waiho*

*Academic Editor*

*PLOS ONE*

*Journal Requirements:*

*Please review your reference list to ensure that it is complete and correct. If you have cited papers that have been retracted, please include the rationale for doing so in the manuscript text, or remove these references and replace them with relevant current references. Any changes to the reference list should be mentioned in the rebuttal letter that accompanies your revised manuscript. If you need to cite a retracted article, indicate the article’s retracted status in the References list and also include a citation and full reference for the retraction notice.*

*Additional Editor Comments (if provided):*

*The experimental design is sound and valid. Statistical analyses were appropriate, after the authors' first revision. I invite the authors to address some minor concerns raised by the reviewers to further improve the quality of the manuscript.*

**

*Reviewers' comments:*

*Reviewer's Responses to Questions*

*
**Comments to the Author**
*

*1. If the authors have adequately addressed your comments raised in a previous round of review and you feel that this manuscript is now acceptable for publication, you may indicate that here to bypass the “Comments to the Author” section, enter your conflict of interest statement in the “Confidential to Editor” section, and submit your "Accept" recommendation.*

*Reviewer #3: (No Response)*

*Reviewer #4: (No Response)*

*2. Is the manuscript technically sound, and do the data support the conclusions?*

*The manuscript must describe a technically sound piece of scientific research with data that supports the conclusions. Experiments must have been conducted rigorously, with appropriate controls, replication, and sample sizes. The conclusions must be drawn appropriately based on the data presented. *

*Reviewer #3: Yes*

*Reviewer #4: Yes*

*3. Has the statistical analysis been performed appropriately and rigorously? *

*Reviewer #3: Yes*

*Reviewer #4: Yes*

*4. Have the authors made all data underlying the findings in their manuscript fully available?*

*The PLOS Data policy requires authors to make all data underlying the findings described in their manuscript fully available without restriction, with rare exception (please refer to the Data Availability Statement in the manuscript PDF file). The data should be provided as part of the manuscript or its supporting information, or deposited to a public repository. For example, in addition to summary statistics, the data points behind means, medians and variance measures should be available. If there are restrictions on publicly sharing data—e.g. participant privacy or use of data from a third party—those must be specified.*

*Reviewer #3: Yes*

*Reviewer #4: Yes*

*5. Is the manuscript presented in an intelligible fashion and written in standard English?*

*PLOS ONE does not copyedit accepted manuscripts, so the language in submitted articles must be clear, correct, and unambiguous. Any typographical or grammatical errors should be corrected at revision, so please note any specific errors here.*

*Reviewer #3: No*

*Reviewer #4: No*

*6. Review Comments to the Author*

*Please use the space provided to explain your answers to the questions above. You may also include additional comments for the author, including concerns about dual publication, research ethics, or publication ethics. (Please upload your review as an attachment if it exceeds 20,000 characters)*

*Reviewer #3: Guppies are a model organism for sexual selection, and there are many empirical studies on mate choice, but in fact most of them are on pre-copulation mate choice, and empirical research has been limited on post-copulation mate choice (cryptic female sperm choice), although its existence has been predicted. The present study is a good study that reported results supporting the existence of cryptic female sperm choice in guppies through a simple problem formulation and carefully manipulated experiments. In addition, Materials and Methods have been properly improved in revised version.*

*I think this MS is suitable for publication in PLoS ONE.*

*I have only a few minor comments.*

*Comments to the authors*

*1) Lines 33-35 in revised MS*

*As reviewer1 mentioned, your results demonstrate that there is a delay in fertilization over days, but your data do not directly showed female cryptic choice.*

*Therefore, you should revise here as you revised in Line 333-335 in revised MS..*

*2) Lines 99-100 in revised MS*

*According to your MS, it seems that the effects of sexual attractiveness of mated males on gestation period in guppies have already revealed in Karino and Sato 2009 (Lines 95-96 [21]).*

*Did you confirm the previous results, or what are the new points for investigation of gestation period in the present study?*

*3) Line 99 in revised MS*

*I think "In this study, we directly tested whether ..." would be more appropriate as you answered to reviewer 2.*

*4) Lines 281-283 in revised MS*

*I agree with your idea. However, you don't explain whether other possibilities remain or not. Therefore, if other possibilities remain, I recommend you to revise the sentence as follows:*

*These results suggest that the difference in gestation periods possibly caused by the difference in timing of fertilization is affected by mate ornamentation.*

*5) Lines 287-288 in revised MS*

*I think the following expression would be more appropriate.*

*"because there are possibilities that some females that paired with drab males failed to fertilize their ova. "*

*Reviewer #4: This is a informative experiment to reveal fundamental reproduction behaviour of guppy in selecting preference mate, which is important to under how female select potential mate and controlling sperm insemination to produce quality offspring. Authors also provided interesting hypothesizes in the introduction section. Overall, MM, Results and Discussion sections support the hypothesizes and addressing objectives of the study. However, conclusion part was not highlighted the aim of the study and hypothesis of the study. Suggest author to summarize the hypothesis of the study at conclusion section. In addition, discussion section (Line 322-330), how female guppy control preference male sperm for fertilisation? this is interesting point to highlight in the study. Suggest authors discuss a little further about this point.*

*language need to be improved, some sentences are difficult to follow especially in the MM & discussion sections.*

*7. PLOS authors have the option to publish the peer review history of their article (what does this mean?). If published, this will include your full peer review and any attached files.*

**

**

*Reviewer #3: No*

*Reviewer #4: **Yes: **Hon Jung Liew*

**

*While revising your submission, please upload your figure files to the Preflight Analysis and Conversion Engine (PACE) digital diagnostic tool, https://pacev2.apexcovantage.com/. PACE helps ensure that figures meet PLOS requirements. To use PACE, you must first register as a user. Registration is free. Then, login and navigate to the UPLOAD tab, where you will find detailed instructions on how to use the tool. If you encounter any issues or have any questions when using PACE, please email PLOS at figures@plos.org. Please note that Supporting Information files do not need this step.*

---

## [Author Response · Author response to Decision Letter 1]

19 Nov 2021

Response to reviewers

We would like to express our appreciation for the reviewers’ suggestions for revising our manuscript. We have improved the manuscript according their comments. The reviewers’ comments are appended hereafter in italics, and our responses are provided in regular font. The line numbers refer to those of the revised manuscript, in which the corresponding changes have been highlighted in the revised manuscript with changes tracked.

Journal Requirements:

>> We have reviewed our reference list and ensured that it is complete and correct. We have not cited papers that have been retracted. 

Additional Editor Comments (if provided):

The experimental design is sound and valid. Statistical analyses were appropriate, after the authors' first revision. I invite the authors to address some minor concerns raised by the reviewers to further improve the quality of the manuscript.

>> Thank you for your comments. We have revised our manuscript as follows: 

Reviewers' comments:

Reviewer's Responses to Questions

Comments to the Author

1. If the authors have adequately addressed your comments raised in a previous round of review and you feel that this manuscript is now acceptable for publication, you may indicate that here to bypass the “Comments to the Author” section, enter your conflict of interest statement in the “Confidential to Editor” section, and submit your "Accept" recommendation.

Reviewer #3: (No Response)

Reviewer #4: (No Response)

2. Is the manuscript technically sound, and do the data support the conclusions?

Reviewer #3: Yes

Reviewer #4: Yes

3. Has the statistical analysis been performed appropriately and rigorously? 

Reviewer #3: Yes

Reviewer #4: Yes

5. Is the manuscript presented in an intelligible fashion and written in standard English?

Reviewer #3: No

Reviewer #4: No

>> The revised manuscript has been edited by a professional language editing service. We have attached a certificate for English editing. 

6. Review Comments to the Author

Reviewer #3: Guppies are a model organism for sexual selection, and there are many empirical studies on mate choice, but in fact most of them are on pre-copulation mate choice, and empirical research has been limited on post-copulation mate choice (cryptic female sperm choice), although its existence has been predicted. The present study is a good study that reported results supporting the existence of cryptic female sperm choice in guppies through a simple problem formulation and carefully manipulated experiments. In addition, Materials and Methods have been properly improved in revised version.

I think this MS is suitable for publication in PLoS ONE.

I have only a few minor comments.

>> Thank you for your comments. We have responded to the comments as follows: 

Comments to the authors

1) Lines 33-35 in revised MS

As reviewer1 mentioned, your results demonstrate that there is a delay in fertilization over days, but your data do not directly showed female cryptic choice.

 Therefore, you should revise here as you revised in Line 333-335 in revised MS..

>>Thank you for pointing this out. We revised the conclusion of the abstract as follows: 

“Our findings show that differences in the timing of fertilization according to attractiveness of the mate increase the opportunity for cryptic female choice and trading up.” (Lines 36-37)

2) Lines 99-100 in revised MS

According to your MS, it seems that the effects of sexual attractiveness of mated males on gestation period in guppies have already revealed in Karino and Sato 2009 (Lines 95-96 [21]).

Did you confirm the previous results, or what are the new points for investigation of gestation period in the present study?

>> In this study, we aimed to determine whether “the difference in gestation period according to male attractiveness in a parturition treatment experiment” and “the difference in the timing of fertilization according to male attractiveness in a laparotomy treatment experiment” correspond. We were afraid that the gestation period or timing of fertilization could be affected by female size or water temperature. In addition, in Karino and Sato (2009), females mated with males only once, but in this study, females were placed in the same tank as males for 24 h. Therefore, we wanted to measure the gestation period and timing of fertilization under the same conditions. 

In addition, control data were needed to support the possibility that unfertilized eggs in the parturition treatment were female manipulation rather than failure in pregnancy.

Based on the above, we examined the female gestation periods in this study. In the revised manuscript, we have changed the end of the introduction as follows: 

“In this study, we directly tested whether the sexual attractiveness of mated males affects the timing of fertilization in female guppies. We observed whether the ova were fertilized by opening their abdomens between 2 and 22 d after pairing with colorful or drab males. Additionally, we measured the gestation period (duration from mating to parturition) between females that mated with colorful males and those that mated with drab males, and compared whether the difference in the timing of fertilization according to male coloration reflected the difference in the gestation period.” (Lines 102-108）

3) Line 99 in revised MS

I think "In this study, we directly tested whether ..." would be more appropriate as you answered to reviewer 2.

>>We revised this sentence according to this comment as follows: 

“In this study, we directly tested whether the sexual attractiveness of mated males affects the timing of fertilization in female guppies.” (Lines 102-103)

4) Lines 281-283 in revised MS

I agree with your idea. However, you don't explain whether other possibilities remain or not. Therefore, if other possibilities remain, I recommend you to revise the sentence as follows:

These results suggest that the difference in gestation periods possibly caused by the difference in timing of fertilization is affected by mate ornamentation.

>> We agree with your comment. We have revised this sentence according to this comment and revision from a professional language editing service. “These results suggest that the difference in gestation periods depending on mate ornamentation is possibly caused by the difference in timing of fertilization.” (Lines 284-285)

5) Lines 287-288 in revised MS

I think the following expression would be more appropriate.

"because there are possibilities that some females that paired with drab males failed to fertilize their ova. "

>> We have revised this sentence according to this comment and revision from a professional language editing service. “because it is possible that some females that paired with drab males failed to fertilize their ova.” (Lines 289-290)

Reviewer #4: This is a informative experiment to reveal fundamental reproduction behaviour of guppy in selecting preference mate, which is important to under how female select potential mate and controlling sperm insemination to produce quality offspring. Authors also provided interesting hypothesizes in the introduction section. Overall, MM, Results and Discussion sections support the hypothesizes and addressing objectives of the study. However, conclusion part was not highlighted the aim of the study and hypothesis of the study. Suggest author to summarize the hypothesis of the study at conclusion section. In addition, discussion section (Line 322-330), how female guppy control preference male sperm for fertilisation? this is interesting point to highlight in the study. Suggest authors discuss a little further about this point.

language need to be improved, some sentences are difficult to follow especially in the MM & discussion sections.

>> Thank you for your helpful comments. We have put a section “Conclusions” to which we have added a summary of the hypothesis of the study (Lines 355-365). In addition, we added to the discussion section a report from a previous study on how females bias paternity toward the second male when they mate sequentially with two males (Lines 329-336).

To improve the language of our manuscript, we have received the service of a professional language editing service and have revised our manuscript.

---

## [Editor Report · Decision Letter 2]

23 Nov 2021

Male coloration affects female gestation period and timing of fertilization in the guppy (*Poecilia reticulata*)

*PONE-D-21-14329R2*

*Dear Dr. Aya Sato,*

*We’re pleased to inform you that your manuscript has been judged scientifically suitable for publication and will be formally accepted for publication once it meets all outstanding technical requirements.*

*Within one week, you’ll receive an e-mail detailing the required amendments. When these have been addressed, you’ll receive a formal acceptance letter and your manuscript will be scheduled for publication.*

*An invoice for payment will follow shortly after the formal acceptance. To ensure an efficient process, please log into Editorial Manager at http://www.editorialmanager.com/pone/, click the 'Update My Information' link at the top of the page, and double check that your user information is up-to-date. If you have any billing related questions, please contact our Author Billing department directly at authorbilling@plos.org.*

*If your institution or institutions have a press office, please notify them about your upcoming paper to help maximize its impact. If they’ll be preparing press materials, please inform our press team as soon as possible -- no later than 48 hours after receiving the formal acceptance. Your manuscript will remain under strict press embargo until 2 pm Eastern Time on the date of publication. For more information, please contact onepress@plos.org.*

*Kind regards,*

*Khor Waiho*

*Academic Editor*

*PLOS ONE*

* *

*Additional Editor Comments (optional):*

* *
---

## [Editor Report · Acceptance letter]

25 Nov 2021

PONE-D-21-14329R2 

Male coloration affects female gestation period and timing of fertilization in the guppy (*Poecilia reticulata*) 

Dear Dr. Sato:

I'm pleased to inform you that your manuscript has been deemed suitable for publication in PLOS ONE. Congratulations! Your manuscript is now with our production department. 

Kind regards, 

on behalf of

Dr. Khor Waiho 

Academic Editor

PLOS ONE